# Slope Stability Analysis to Correlate Shear Strength with Slope Angle and Shear Stress by Considering Saturated and Unsaturated Seismic Conditions

**Muhammad Israr Khan** *[ID] **and Shuhong Wang**

School of Resources and Civil Engineering, Northeastern University, Shenyang 110819, China; wangshuhong@mail.neu.edu.cn
* Correspondence: m.israr@abasyn.edu.pk; Tel.: +86-131-4980-8192

**Featured Application: Landsliding is a big problem faced by people all over the world. This paper provides a solution to understand the soil behavior and hence to design the slopes in such a way that the risk of landsliding is in the minimum range.**

**Abstract:** Assessment and analysis of soil slope stability is an important part of geotechnical engineering at all times. This paper examines the assessment of soil slope stability in fine-grained soils. The effect of change in shear strength ($\tau$), shear stress ($\sigma$) and slope angle ($\beta$) on the factor of safety has been studied. It correlates shear strength with slope angle and shear stress by considering the horizontal seismic coefficients in both saturated and unsaturated conditions. The slope failure surface was considered a circular slip surface. Statistical package for social sciences (SPSS) and Slide, numerical modeling software and limit equilibrium slope stability analysis software, respectively, are used to find out the correlations between the three basic parameters. The slope angle varied from 70 to 88 degrees, which are the most critical values for slope angles, and a total of 200 analyses were performed. $\tau$, $\beta$ and $\sigma$ are correlated, and the correlations are provided in the results section. The results indicate that the correlations developed between the parameters have a very close relationship. The applicability of the developed equations is above 99%. These correlations are applicable in any type of soil slope stability analysis, where the value of shear strength and factor of safety is required with the variation of slope angle and shear stress.

**Keywords:** slope stability analysis; shear strength; slope angle; shear stress; correlations

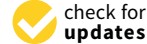



## 1. Introduction

A slope's soil, separated by common type of geologic body and slope stability analysis, is very important to know regarding the safety of the slope. Rain and earthquakes are two major causes of landslides. Generally, it is necessary to monitor rainfall and earthquakes at the same time to assess the stability of the slope. Many useful and detailed analyses have already been published regarding landslides due to rainfall or earthquakes [1–4]. Landsliding of artificial or natural soil slopes normally occurs during or after strong earthquakes. These landsliding events depend on earthquake loading and geological conditions. Compared to the damage caused by the earthquake itself, it has been observed that earthquake-induced landslides cause more damage to infrastructure. Dynamic slope stability analysis is broadly used to compute and analyze the seismic stability of manmade and natural slopes [5]. Pore-water pressure is increased after rainfall and hence lowers the active pressure and reduces the soil's shear strength [6]. Field and laboratory experiments were performed to study the rainfall-induced soil slope failures [7]. Another researcher examined laboratory, field tests and the study of specific, large and accessible data based on 49 different landslides caused by the worst rainfall events in northern Australia and gave an indication of possible earthquake-induced landslide sites in the future [7]. Numerous quantitative studies have

been conducted to investigate infiltration issues [8,9]. Some other researchers have also studied the limits on the stability analysis of the soil slope under rainfall [10]. In the case of landslides caused by earthquakes, the dynamic performance of slopes depends on factors such as slope geometry, magnitude, and seismic signals [11–13]. The earthquake-induced landslide susceptibility, earthquake-affected landslide spatial distribution and dynamic slope failure mechanisms have been studied [14–18]. The difference in dynamic response between the counter-bedding slope and bedding slope by using the shaking table test was checked, and it was found that the displacement of the bedding slope surface was more than that of the counter-bedding slope surface [19]. In Turkey, a study was conducted to examine the multi-dimensional mechanism of a landslide induced by earth-shaking using dynamic numerical modeling as well as the limit equilibrium method. The study concluded that a multi-dimensional roto-translational mechanism is responsible for the landslide mass into sub-masses and hence overcoming the characteristic period related to the landslide length [20]. Considering the combined effect of rain and earthquakes, laboratory experiments in controlled conditions are performed where most of the uncertainty of actual conditions is absent or less widespread. For example, a researcher [21] conducted an experimental analysis to study the combined effect of rainfall and earthquakes on the loess slopes. This is the only example that was found that combined both of these phenomena. The effect of weak layers on the failure of the soil slope due to earthquakes was not considered. It was also found that the weak soil layers can significantly weaken the earthquake amplitude, in the shaking table test [22]. Critical acceleration is a natural feature of the slope, and it determines the stability of the slope under the action of an earthquake. The critical acceleration model is a key component of the regional earthquake risk assessment. Therefore, the purpose of this paper is to highlight the impact of various types of acceleration on the assessment of the potential landslide caused by earthquakes. Traditionally, Newmark's sensitive acceleration model is often used to assess the potential cause of landslides. This method requires the soil properties as input and to estimate the shear strength of soil to find out the factor of safety. The factor of safety plays an important role in the design of an embankment. If this value is not calculated with great care, then the designed slope might fail before the end of its designed lifetime. Hence, great care must be used while computing the factor of slope safety. Dynamic analysis is also very important to consider while designing any slope. Investigating earthquake response and severe failure of soil slopes facing earthquakes and rainfall, two types of slopes having saturated and unsaturated seismic conditions are considered, and correlations are developed between shear strength with slope angle and shear stress.

Failure of natural slopes (landslides) and manmade slopes have led to significant death and destruction, economic losses and environmental damages. Some failures are sudden and catastrophic, while others are time-consuming and offer some time to escape. Some failures are widespread, and some are local. Geotechnical engineers should pay close attention to geology, surface drainage, groundwater and the shear strength of the soil to assess slope stability. Most of these failures are due to the shear strength failure of soil. Figure 1 presents a simple example of soil shear failure (soil slippage) in a remote area of Washington (USA) during a rainy season and hence caused a huge landslide. Fortunately, no deaths were reported.

It is observed that shear strength has a very close connection with slope angle, which is also called the angle of repose ($\beta$). The results are always different for seismic analysis and non-seismic analysis. The reliable numerical modeling and simulation of any soil slope landslide process contributes to the knowledge about the occurrence of slope failure in seismically active zones. To achieve this, a simple shear strength soil slope model is taken into account to describe the shear strength variation in a seismically active landslide process. After running the analysis, the proposed slope model develops a correlation between the shear strength with shear stress and slope angle.

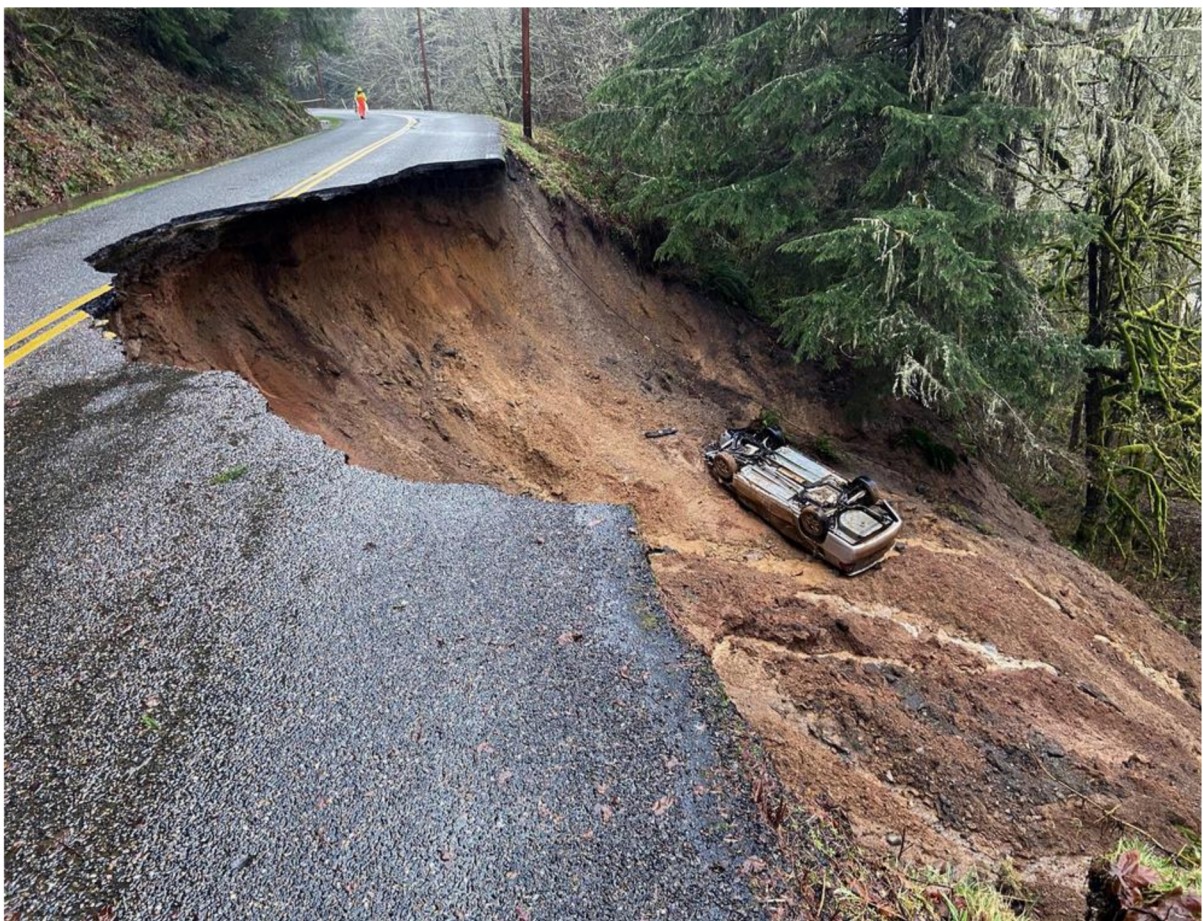

**Figure 1.** Landslide in Washington—18 February 2020 [23].

## 2. Materials and Methods

The past ten years have seen an increase in problem-solving through numerical modeling and analysis in the field of engineering sciences. The popularity and versatility of these methods have greatly increased due to the availability of high-speed digital computers. This paper is also based on the numerical modeling of a soil slope model.

*Sampling and Testing*

Mozishan Park in Shenyang city, China, is the site from where the soil sample was collected. This whole area consists of many slopes, and landsliding happens occasionally. The soil at this specific site consists of clay and clayey sand. Soil samples were collected with the help of a drilling machine having 6 inches diameter. Figure 2 presents the site area (Coordinates: 41.666943, 123.477476).

A total of ten boreholes were drilled to collect samples at almost every corner and central position of the site. After sample collection, all necessary and required laboratory tests were performed, such as the triaxial test, moisture content test and direct shear test. In situ direct shear tests and triaxial tests are conducted in the laboratory to find out the material properties. The Mohr–Coulomb envelope criteria are used to compute the cohesion and friction of soil. The general form of Mohr–Coulomb criteria is shown in Equation (1).

$$\tau = c + \sigma \tan \Phi \tag{1}$$

Table 1 presents all the material properties achieved in the experimental analysis.

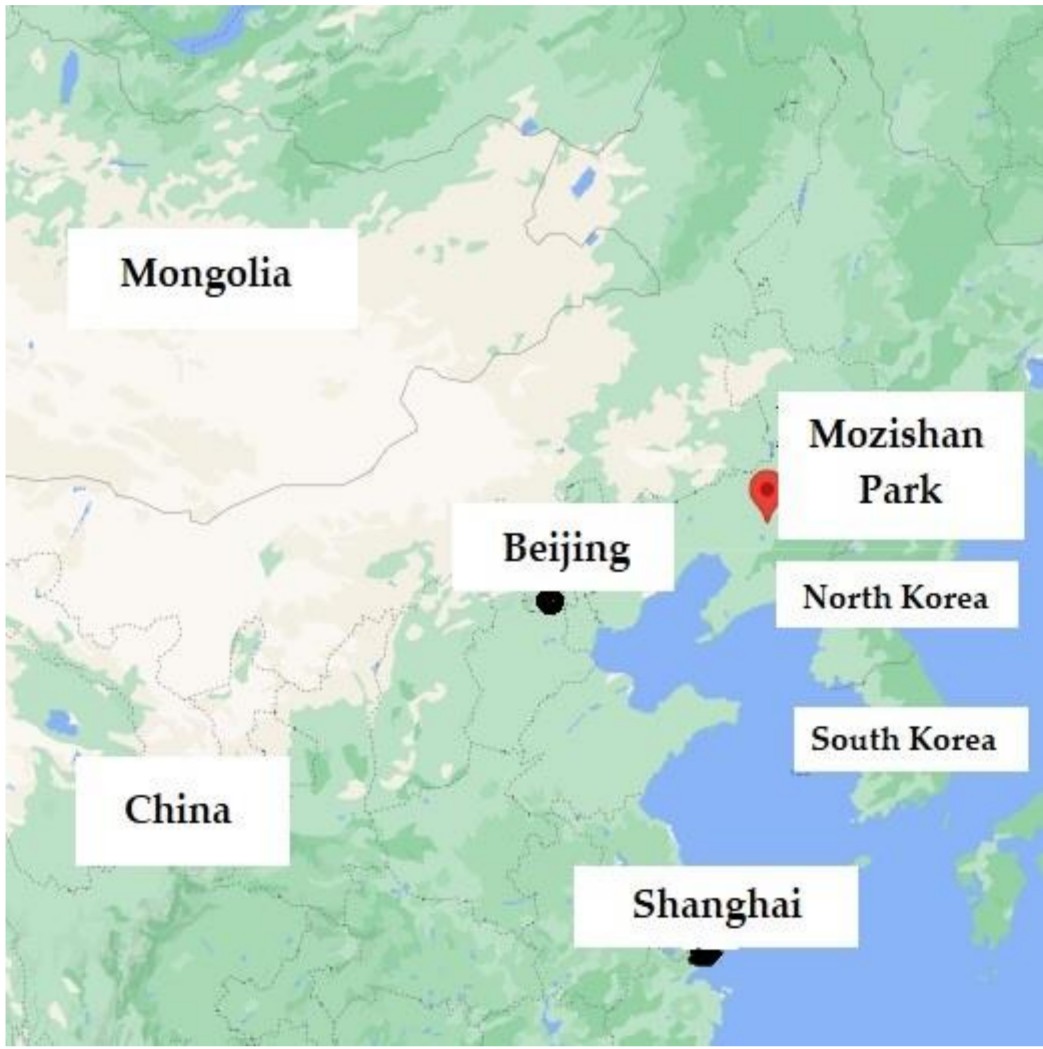

**Figure 2.** Mozishan Park site area [24].

**Table 1.** Material properties.

| Material Number | Cohesion (kPa) | Friction Angle (Degrees) | Unit Weight (kN/m$^3$) |
|:---:|:---:|:---:|:---:|
| 1 | 25 | 36 | 16.5 |
| 2 | 25.8 | 35.5 | 16.2 |
| 3 | 26 | 35 | 15.9 |
| 4 | 26.3 | 34.6 | 15.5 |
| 5 | 26.9 | 34.2 | 15.1 |
| 6 | 27.4 | 33.7 | 14.8 |
| 7 | 28 | 33.2 | 14.4 |
| 8 | 28.5 | 31.8 | 14 |
| 9 | 29 | 31.4 | 13.6 |
| 10 | 29.4 | 30.9 | 13.1 |

According to the actual site condition, a 2D slope was modeled in Slide (2D limit equilibrium software) for further analysis. The coordinates of the slope are mentioned in Figure 3. System international (SI) units were used in this analysis. Figure 3 presents the slope model, including the coordinates of each corner that were considered in this analysis. Figure 4 is the methodology flowchart.

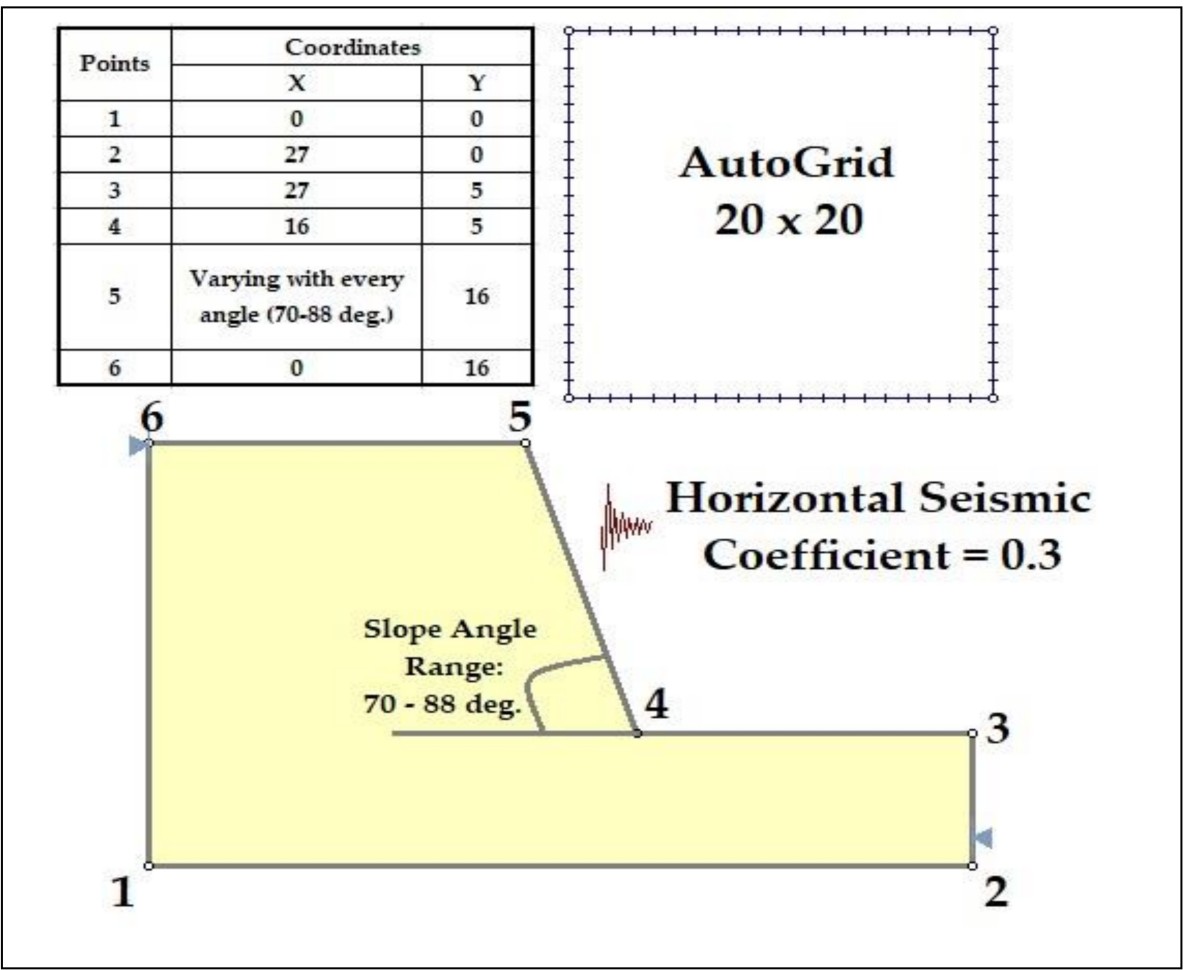

**Figure 3.** Slope model with coordinates.

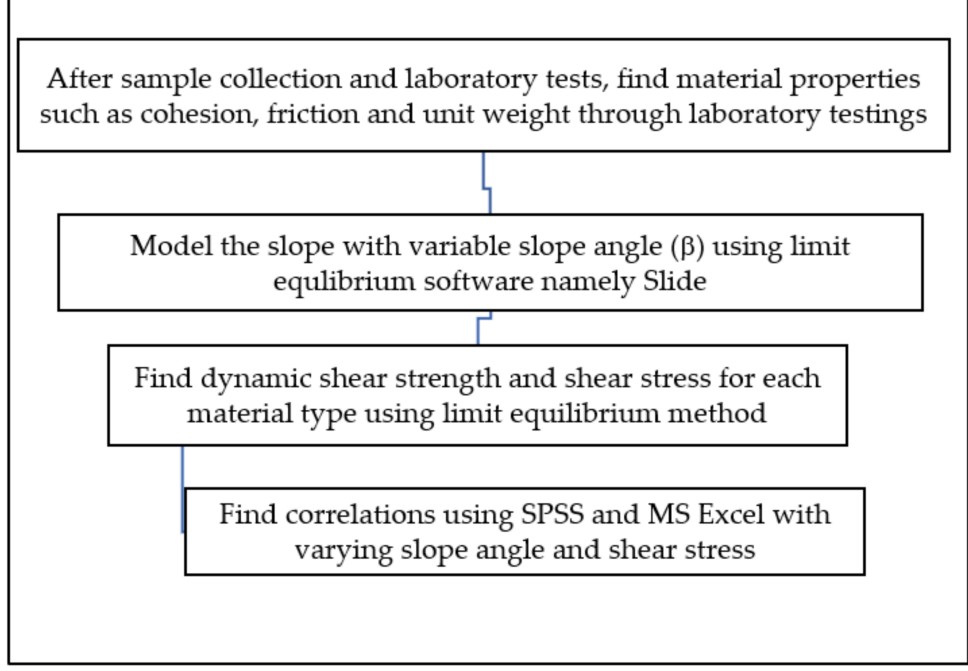

**Figure 4.** Methodology flowchart.

Comparative modeling of soil behavior with the influence of different soil parameters is an important research area for and many studies [25–31] have been conducted. In the case of weak soil strength, which needs improvement, there are many methods to improve the soil shear strength. For instance, diatomite species increased both residual and peak shear strength of soil [32]. Similarly, the occurrence of landslides is a geohazard problem that affects many regions of the world [33]. Landslides may occur due to natural events such as earthquakes, rain, volcanic eruption and also by manmade changes to the ground and slopes. Most landslide events occur due to the low value of shear strength.

By using linear regression analysis, prediction equations have been developed in this research paper. Statistical package for the social sciences (SPSS) software is used to compute the linear correlations between shear strength of soil, slope angle and shear stress. SPSS is normally used for analyzing complex mathematical data to find correlations between different variables. It can perform both linear and non-linear regression analyses. In this paper, the Chinese site is considered, and the material properties are computed by different laboratory tests such as Atterberg limits, sieve analysis, triaxial test and moisture content test. Table 1 presents the material properties. $\beta$ range is kept between 70 and 88 degrees. This is because the most critical $\beta$ value is in the range of 60 to 90 degrees. A total of 10 different material types are considered out of the 10 boreholes. After all necessary laboratory testing, the material properties ranges came out to be:

Unit weight: 13.10 to 16.50 (kN/m$^3$), cohesion: 25.00 to 29.00 (kPa), friction angle: 30 to 36 degrees and slope angle: 70 to 88 degrees. Out of the many slope stability analysis methods, the Fellenius analysis method is considered in this work as it gives the most optimum results. It also gives a smaller value for the slope factor of safety. The limit equilibrium software, namely Slide, offers all analysis methods such as Bishop simplified, Corps of Engineers # 1, Corps of Engineers # 2, GLE/Morgenstern–Price, Janbu simplified, Janbu corrected, Lowe–Karafiath, Ordinary/Fellenius and Spencer method. Out of all these methods, it is observed that the Fellenius method gives a lower slope factor of safety. Considering a lower value for slope factor of safety while designing a slope provides a higher factor of safety. Therefore, the Fellenius method is preferred in this analysis. Figure 5a,b presents the assumed slices and forces acting on each slice, respectively.

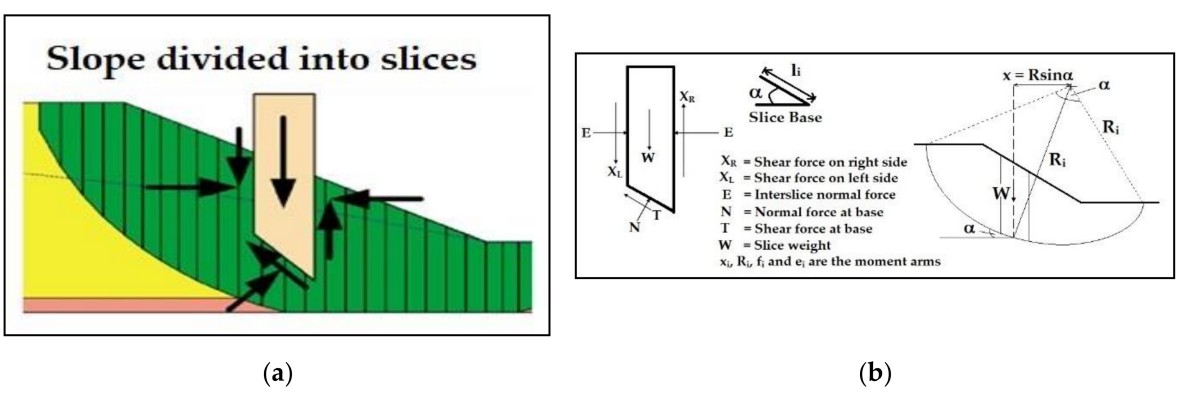

| (**a**) | (**b**) |

**Figure 5.** (**a**). Assumed slices in the method of slices. (**b**) Forces acting on each slice.

The force applied to each slice is obtained by analyzing the mechanical balance of each component. Each slice is considered separate, and the interaction between the slices is ignored because the resultant strengths are the same as the base of each slice. However, Newton's third law is not satisfied in this method because, in general, the effects on the left and right of the slices are not of the same magnitude and are not collinear. In this method, the resultant horizontal and vertical equilibrium equations are as follows:

$$\sum F_v = W - N\cos\alpha - T\sin\alpha = 0 \tag{2}$$

$$\sum F_h = kW + N\sin\alpha - T\cos\alpha = 0 \tag{3}$$

where k represents a factor that computes the horizontal force with depth.

$$N = W\cos\alpha - kW\sin\alpha \tag{4}$$

Similarly, moment equation in this method is:

$$\sum M = \sum(W_i x_i - T_i R_i - N_i f_i - kW_i e_i) = 0 \tag{5}$$

where i is the slice index while $x_i$, $R_i$, $f_i$ and $e_i$ are the moment arms.

Putting the value of normal force in Equation (5) results in:

$$\sum_i T_i R_i = \sum[W_i x_i - (W_i \cos\alpha_i - kW_i \sin\alpha_i)f_i - kW_i e_i] \tag{6}$$

Using Terzaghi's theory, the equation could be written in moment form as:

$$\sum_i = \tau l_i R_i = l_i R_i \sigma'_i \tan\Phi' + l_i R_i c' = R_i (N_i - u_i l_i) \tan\Phi' + l_i R_i c' \tag{7}$$

The factor of safety equation in the method of slices is:

$$\text{Factor of safety} = \frac{\sum i = \tau li\, Ri}{\sum i\, Ti\, Ri} \tag{8}$$

Numerical modeling provides a limited solution to problems that cannot be solved by conventional methods, e.g., complex geometry, material anisotropy, linear behavior and in situ pressures related models. Numerical analysis allows for soil fluctuations and failures, pore pressure modeling, unequal intensity, dynamic loading and testing parameters for variable parameters etc.

### 3. Results

*3.1. Saturated Seismic Analysis*

Figure 6 presents the slope model considered in this phase. The vertical seismic coefficient is neglected as it is very low or zero in most cases. The horizontal seismic load was applied using the seismic loading option in the Slide software. The seismic coefficients are dimensionless coefficients that represent the maximum earthquake acceleration as a fraction of the acceleration due to gravity. Typical values are in the range of 0.1 to 0.3. In this paper, the maximum value for the horizontal seismic coefficient is considered as 0.3.

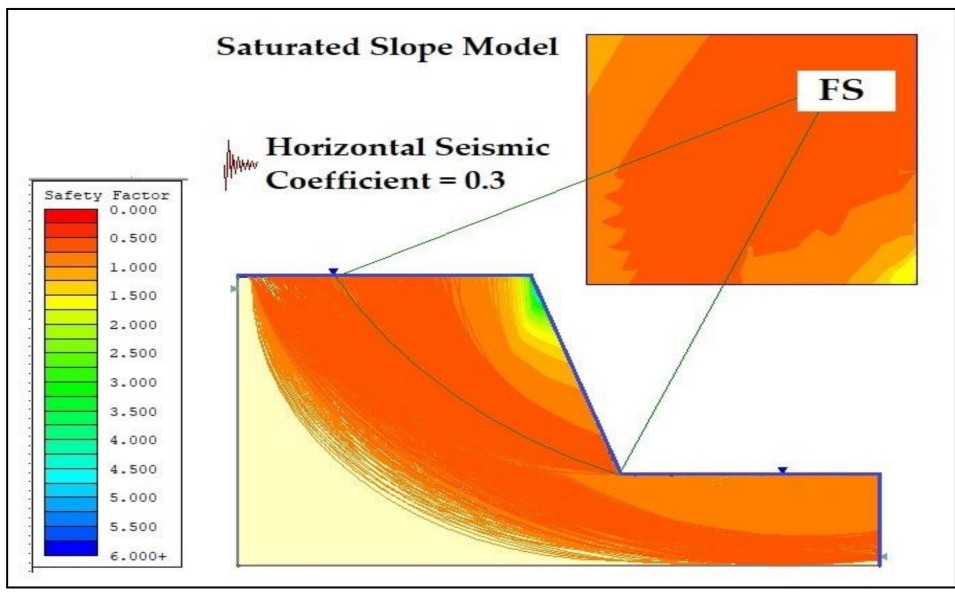

**Figure 6.** Slope model—saturated seismic case.

The coordinates (m) of the slope are 0,0; 27,0; 27,5; 16,5, these vary with the slope angle, 16 and 16, 0. The model's coordinates are in meters. In the fifth coordinate, variations in the slope angle mean that this coordinate value changes from 70 to 88 degrees. For all of the 200 analyses, this coordinate has a unique value. This slope is analyzed dynamically by applying the maximum horizontal seismic coefficient of 0.3. The water table is applied at the upper surface of the slope. This analysis is performed in the total saturated condition. Normally slopes are subjected to rain and water, such as rivers, at the bottom. Therefore, the more accurate results will be considered as those which consider all of the environmental conditions. Studies regarding the rain effects on landsliding are one of the most challenging fields in geotechnical engineering. It is observed that an increase in the water table will also increase water pore pressure, which pushes the soil particles away from each other, and hence reduces the stress between the particles resulting in soil slope failure. Two such papers [34,35] may be used as a guide to finding the adequate literature on the most used or the most advanced approaches followed in every step of the procedure for defining reliable rainfall thresholds. Figure 7 presents the factor of safety values achieved in the case of Material 1 for slope angle ranges from 70 to 88 degrees, respectively.

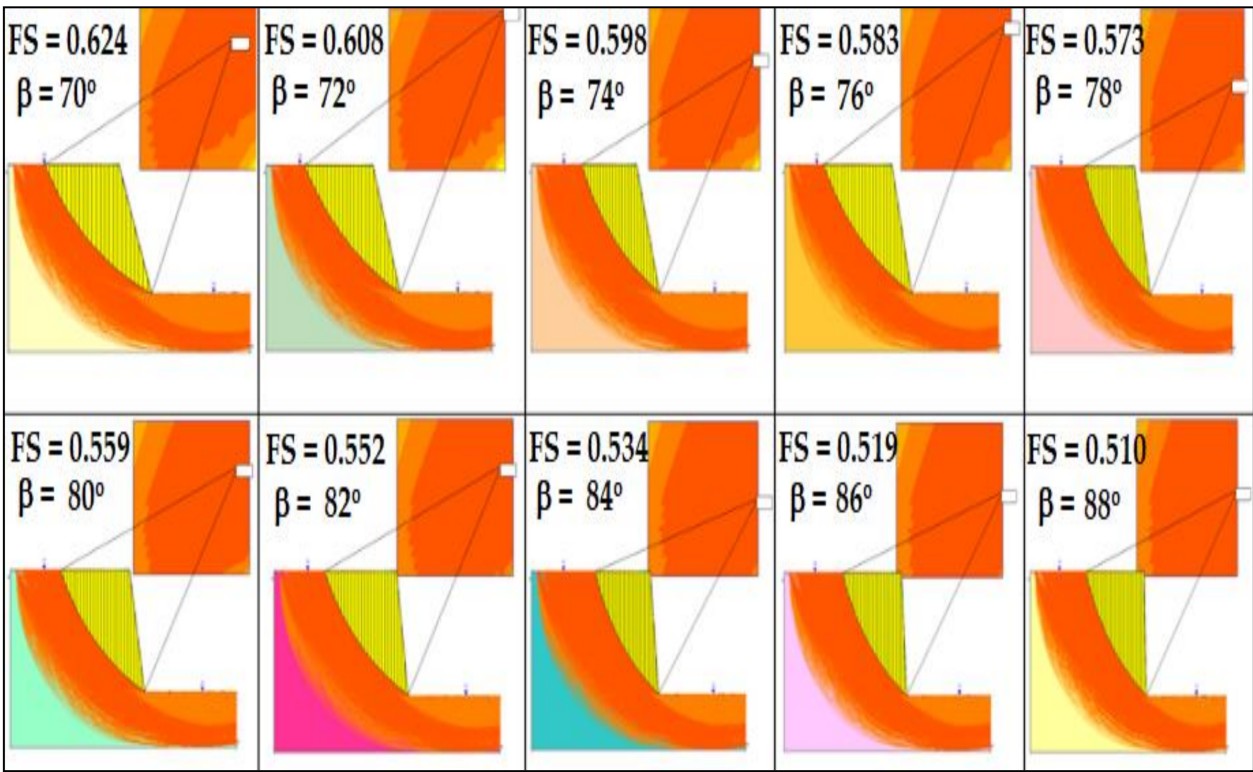

**Figure 7.** Factor of safety values in the case of Material 1—saturated seismic case.

It is clear from Figure 7 that with increasing $\beta$, FS will decrease at a constant rate. The FS values for Material 1 in the case of a saturated seismic condition are 0.624, 0.608, 0.598, 0.583, 0.573, 0.559, 0.552, 0.534, 0.519 and 0.510, respectively. A similar analysis was conducted for all material types, and the summary is provided in Tables 2 and 3 along with FS values.

**Table 2.** τ and σ in the case of a saturated seismic analysis—Materials 1 to 5.

| Slope Angle | Material 1 | | | Material 2 | | | Material 3 | | | Material 4 | | | Material 5 | | |
|---|---|---|---|---|---|---|---|---|---|---|---|---|---|---|---|
| | τ (kPa) | σ (kPa) | FS | τ (kPa) | σ (kPa) | FS | τ (kPa) | σ (kPa) | FS | τ (kPa) | σ (kPa) | FS | τ (kPa) | σ (kPa) | FS |
| 70 | 40.68 | 65.24 | 0.624 | 40.11 | 62.68 | 0.640 | 38.99 | 60.47 | 0.646 | 37.69 | 57.68 | 0.654 | 38.05 | 56.82 | 0.670 |
| 72 | 38.58 | 63.46 | 0.608 | 38.17 | 61.07 | 0.625 | 37.20 | 59.01 | 0.630 | 36.09 | 56.40 | 0.640 | 35.33 | 53.77 | 0.657 |
| 74 | 35.39 | 59.20 | 0.598 | 38.96 | 63.33 | 0.615 | 37.88 | 61.09 | 0.620 | 36.65 | 58.26 | 0.629 | 38.65 | 60.01 | 0.644 |
| 76 | 40.21 | 69.02 | 0.583 | 39.63 | 66.24 | 0.598 | 38.50 | 63.85 | 0.603 | 37.20 | 60.84 | 0.611 | 36.24 | 57.80 | 0.627 |
| 78 | 36.27 | 63.30 | 0.573 | 40.83 | 69.13 | 0.591 | 40.20 | 67.59 | 0.595 | 39.39 | 65.44 | 0.602 | 38.28 | 62.19 | 0.616 |
| 80 | 40.17 | 71.86 | 0.559 | 39.56 | 68.82 | 0.575 | 39.88 | 68.80 | 0.580 | 38.45 | 65.44 | 0.588 | 37.37 | 62.02 | 0.603 |
| 82 | 41.21 | 74.72 | 0.552 | 40.57 | 71.59 | 0.567 | 39.38 | 68.93 | 0.571 | 38.74 | 66.87 | 0.579 | 38.83 | 65.43 | 0.593 |
| 84 | 35.82 | 67.09 | 0.534 | 35.46 | 64.18 | 0.553 | 40.04 | 71.71 | 0.558 | 38.59 | 68.09 | 0.567 | 37.50 | 64.42 | 0.582 |
| 86 | 38.92 | 75.04 | 0.519 | 38.38 | 71.68 | 0.535 | 37.29 | 68.86 | 0.541 | 36.02 | 65.34 | 0.551 | 35.09 | 61.79 | 0.568 |
| 88 | 40.30 | 79.05 | 0.510 | 39.67 | 75.44 | 0.526 | 38.49 | 72.42 | 0.531 | 37.12 | 68.67 | 0.541 | 36.10 | 64.86 | 0.557 |

**Table 3.** τ and σ in the case of a saturated seismic analysis—Materials 6 to 10.

| Slope Angle | Material 6 | | | Material 7 | | | Material 8 | | | Material 9 | | | Material 10 | | |
|---|---|---|---|---|---|---|---|---|---|---|---|---|---|---|---|
| | τ (kPa) | σ (kPa) | FS | τ (kPa) | σ (kPa) | FS | τ (kPa) | σ (kPa) | FS | τ (kPa) | σ (kPa) | FS | τ (kPa) | σ (kPa) | FS |
| 70 | 37.33 | 54.69 | 0.683 | 36.42 | 52.02 | 0.700 | 37.12 | 52.01 | 0.714 | 38.38 | 52.61 | 0.730 | 39.64 | 53.59 | 0.740 |
| 72 | 34.77 | 51.78 | 0.672 | 38.91 | 56.55 | 0.688 | 37.36 | 53.37 | 0.700 | 36.30 | 50.67 | 0.716 | 39.61 | 54.01 | 0.733 |
| 74 | 37.83 | 57.61 | 0.657 | 36.85 | 54.70 | 0.674 | 37.12 | 54.08 | 0.686 | 36.13 | 51.45 | 0.702 | 36.05 | 50.18 | 0.718 |
| 76 | 35.54 | 55.51 | 0.640 | 34.68 | 52.70 | 0.658 | 33.73 | 50.02 | 0.674 | 36.08 | 52.21 | 0.691 | 34.61 | 48.88 | 0.708 |
| 78 | 37.46 | 59.72 | 0.627 | 36.42 | 56.64 | 0.643 | 35.10 | 53.44 | 0.657 | 34.14 | 50.68 | 0.674 | 32.78 | 47.39 | 0.692 |
| 80 | 36.58 | 59.44 | 0.615 | 35.56 | 56.23 | 0.632 | 35.90 | 55.41 | 0.648 | 34.85 | 52.47 | 0.664 | 33.38 | 48.99 | 0.681 |
| 82 | 38.94 | 64.41 | 0.605 | 38.48 | 62.14 | 0.619 | 36.86 | 58.37 | 0.632 | 35.69 | 55.16 | 0.647 | 34.07 | 51.36 | 0.663 |
| 84 | 36.68 | 61.65 | 0.595 | 37.88 | 62.01 | 0.611 | 37.55 | 60.32 | 0.623 | 36.99 | 58.11 | 0.637 | 35.18 | 54.04 | 0.651 |
| 86 | 34.42 | 59.12 | 0.582 | 38.52 | 64.56 | 0.597 | 36.88 | 60.52 | 0.609 | 39.60 | 64.77 | 0.611 | 35.47 | 55.35 | 0.641 |
| 88 | 35.35 | 62.00 | 0.570 | 34.39 | 58.47 | 0.588 | 39.36 | 65.51 | 0.601 | 37.93 | 61.78 | 0.614 | 36.01 | 57.38 | 0.628 |

Tables 2 and 3 present the shear strength (τ) and shear stress (σ) values for Materials 1 to 5 and 6 to 10, respectively, in the case of a saturated seismic condition. While Table 4 presents the correlations developed between τ, σ and β for all the ten material types.

**Table 4.** Correlations in the case of clay—Saturated seismic case.

| Material Number | Shear Strength (kPa) |
|---|---|
| 1 | $34.572 - 0.434*\beta + 0.559*\sigma$ <br> $R^2 = 99.4\%$ |
| 2 | $33.791 - 0.412*\beta + 0.562*\sigma$ <br> $R^2 = 98.9\%$ |
| 3 | $31.792 - 0.455*\beta + 0.648*\sigma$ <br> $R^2 = 99.2\%$ |
| 4 | $30.325 - 0.415*\beta + 0.633*\sigma$ <br> $R^2 = 99.6\%$ |
| 5 | $29.162 - 0.388*\beta + 0.634*\sigma$ <br> $R^2 = 99.8\%$ |
| 6 | $28.600 - 0.355*\beta + 0.613*\sigma$ <br> $R^2 = 99.8\%$ |
| 7 | $28.849 - 0.353*\beta + 0.622*\sigma$ <br> $R^2 = 99.5\%$ |
| 8 | $28.884 - 0.360*\beta + 0.645*\sigma$ <br> $R^2 = 99.9\%$ |
| 9 | $30.500 - 0.325*\beta + 0.579*\sigma$ <br> $R^2 = 99.1\%$ |
| 10 | $28.343 - 0.349*\beta + 0.670*\sigma$ <br> $R^2 = 99.9\%$ |

Figure 8a presents the shear strength and shear stress graphs achieved in the case of Material 1 having β equals 70 and 88 degrees in a saturated condition. Figure 8b presents shear strength and shear stress graphs in the case of Material 1 with β equals 78 and 88 degrees, respectively.

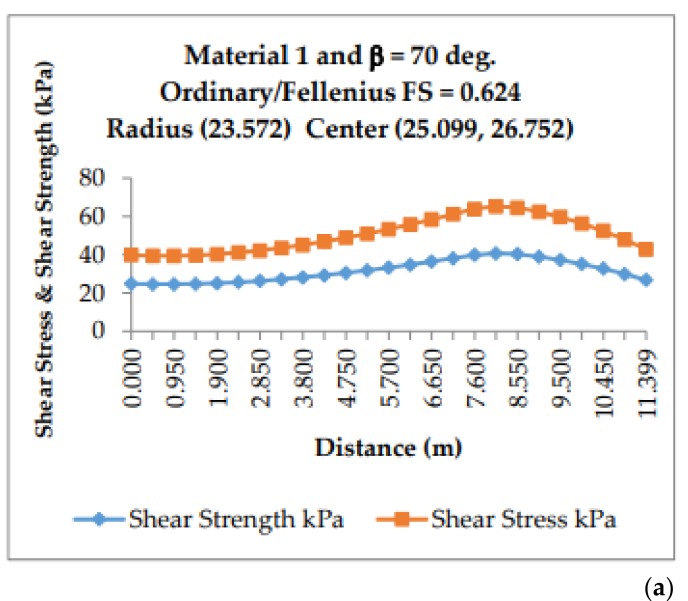

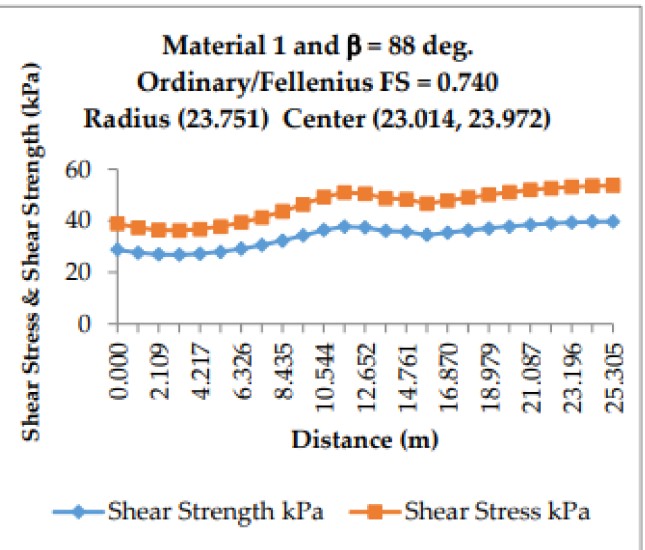

(a)

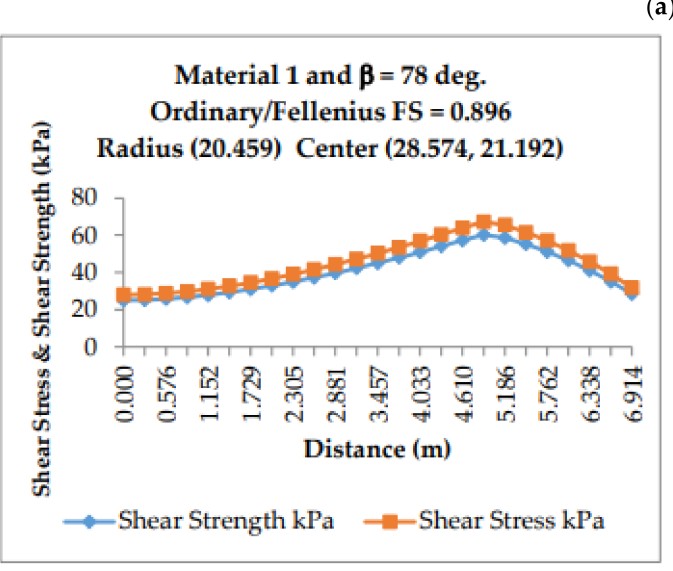

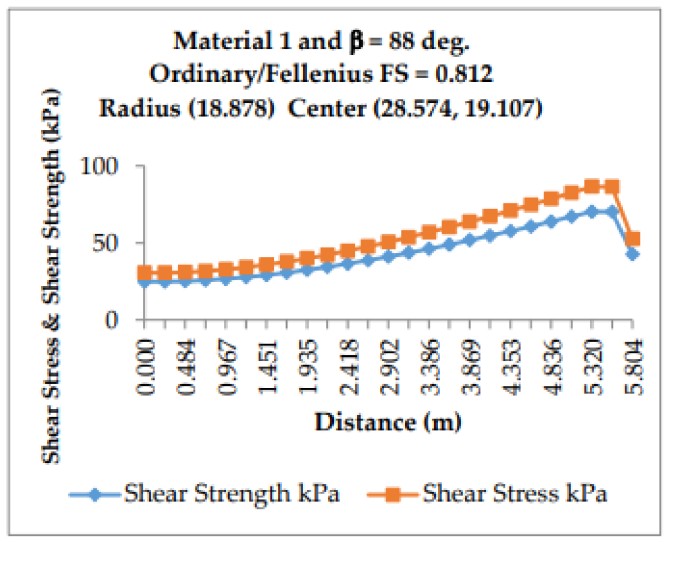

(b)

**Figure 8.** (**a**). Shear strength and shear stress graphs in the case of Material 1 and 10 in a saturated case. (**b**). Shear strength and shear stress graphs in the case of Material 5 and 1 in an unsaturated case.

Figure 8a,b presents a clear indication of shear strength dependency upon shear stress. The value increases till the failure point is reached and then declines. With the variation of β, shear stress changes and hence shear strength also changes at a constant rate. All these parameters are closely related.

In all these correlations, the applicability value $R^2$ is not less than 99% in any case. This presents the equation's reliability and applicability as very high. From Table 5, the final correlation is calculated by computing the average value, which is:

$$\tau \text{ (kPa)} = 30.482 - 0.385\beta \text{ (deg.)} + 0.617\sigma \text{ (kPa)} \tag{9}$$

and the average $R^2$ comes out to be 99.5%.

**Table 5.** τ and σ in the case of an unsaturated seismic analysis—Materials 1 to 5.

| Slope Angle | Material 1 | | | Material 2 | | | Material 3 | | | Material 4 | | | Material 5 | | |
|---|---|---|---|---|---|---|---|---|---|---|---|---|---|---|---|
| | τ (kPa) | σ (kPa) | FS | τ (kPa) | σ (kPa) | FS | τ (kPa) | σ (kPa) | FS | τ (kPa) | σ (kPa) | FS | τ (kPa) | σ (kPa) | FS |
| 70 | 62.49 | 63.71 | 0.981 | 61.94 | 61.68 | 1.004 | 63.15 | 62.33 | 1.013 | 61.98 | 60.12 | 1.031 | 63.44 | 60.04 | 1.057 |
| 72 | 61.31 | 64.24 | 0.954 | 60.80 | 62.16 | 0.978 | 59.73 | 60.42 | 0.988 | 58.69 | 58.25 | 1.007 | 57.99 | 55.99 | 1.036 |
| 74 | 64.41 | 68.43 | 0.941 | 63.78 | 66.17 | 0.964 | 65.27 | 67.04 | 0.974 | 64.02 | 64.61 | 0.991 | 66.14 | 65.06 | 1.017 |
| 76 | 59.89 | 65.12 | 0.920 | 59.43 | 62.89 | 0.945 | 58.41 | 61.07 | 0.956 | 57.42 | 58.80 | 0.977 | 65.39 | 65.16 | 1.004 |
| 78 | 60.24 | 67.24 | 0.896 | 59.77 | 64.92 | 0.921 | 58.73 | 63.02 | 0.932 | 57.73 | 60.66 | 0.952 | 65.81 | 67.15 | 0.980 |
| 80 | 67.21 | 76.60 | 0.877 | 66.49 | 73.93 | 0.899 | 65.20 | 71.76 | 0.909 | 63.95 | 69.05 | 0.926 | 63.03 | 66.22 | 0.952 |
| 82 | 70.91 | 81.90 | 0.866 | 70.06 | 79.00 | 0.887 | 68.64 | 76.65 | 0.896 | 67.25 | 73.73 | 0.912 | 66.20 | 70.66 | 0.937 |
| 84 | 69.13 | 81.45 | 0.849 | 68.33 | 78.46 | 0.871 | 66.98 | 76.05 | 0.881 | 65.66 | 73.08 | 0.899 | 72.99 | 79.11 | 0.923 |
| 86 | 67.18 | 81.41 | 0.825 | 66.46 | 78.33 | 0.848 | 65.17 | 75.87 | 0.859 | 63.92 | 72.85 | 0.877 | 63.00 | 69.67 | 0.904 |
| 88 | 70.24 | 86.52 | 0.812 | 69.41 | 83.25 | 0.834 | 68.02 | 80.63 | 0.844 | 66.65 | 77.41 | 0.861 | 65.63 | 74.02 | 0.887 |

### 3.2. Unsaturated Seismic Analysis

Another case was the analysis of the same slope and same material properties in the case of an unsaturated condition. In this section, the same method (Fellenius) is followed for slope stability analysis, computing the shear strength, shear stress and factor of safety. The measurement of soil suction is performed through the Slide software. It is measured in terms of the height of the water column (h) suspended in the soil. Figure 9 presents the slope model considered in this phase. Figure 10 presents the factor of safety achieved in the case of Material 1 for slope angles ranging from 70 to 88 degrees, respectively.

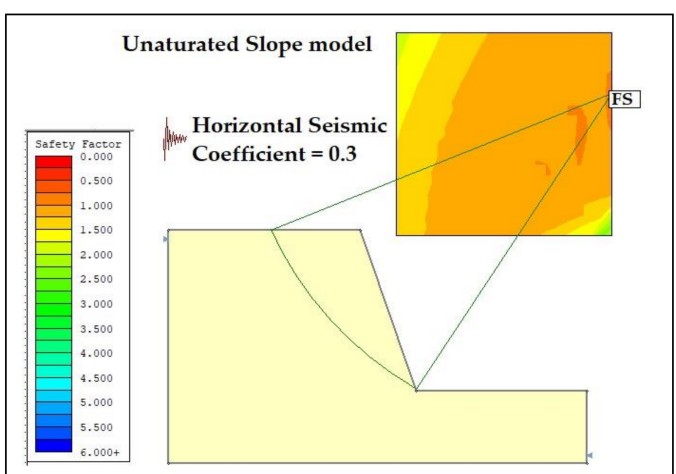

**Figure 9.** Slope model—unsaturated seismic case.

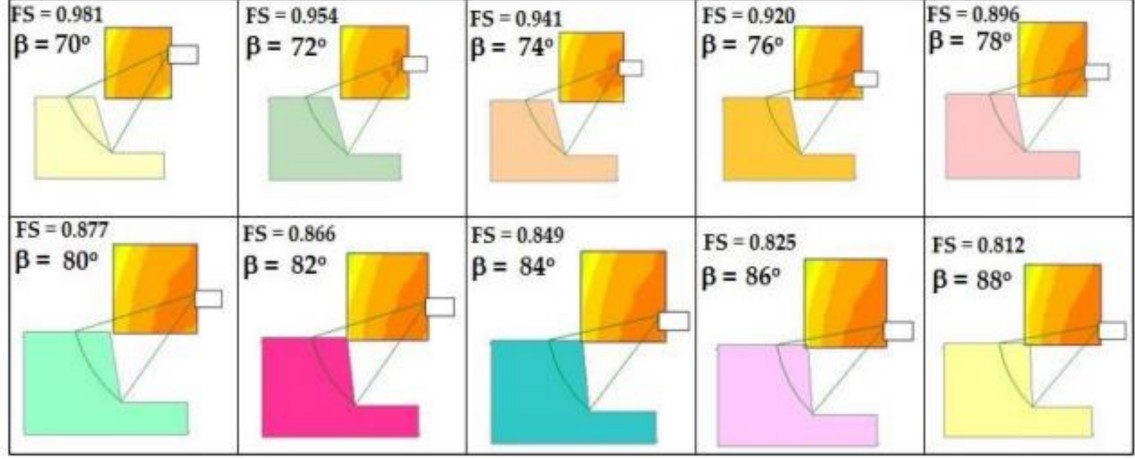

**Figure 10.** Factor of safety values in the case of Material 1—unsaturated seismic case.

The FS values for Material 1 in the case of a saturated seismic condition are 0.981, 0.954, 0.941, 0.920, 0.896, 0.877, 0.866, 0.849, 0.825 and 0.812, respectively. Similar analyses, considering the material properties mentioned in Table 1, were performed for all material types, and its summary is provided in Tables 5 and 6. All the input values were analyzed in Slide and Tables 5 and 6 are achieved, which presents the shear strength ($\tau$) and shear stress ($\sigma$) along with FS values for Materials 1 to 5 and 6 to 10, respectively, in the case of an unsaturated seismic condition. A total number of 100 analyses were performed in this case.

**Table 6.** $\tau$ and $\sigma$ in the case of an unsaturated seismic analysis—Materials 6 to 10.

| Slope Angle | Material 6 | | | Material 7 | | | Material 8 | | | Material 9 | | | Material 10 | | |
|---|---|---|---|---|---|---|---|---|---|---|---|---|---|---|---|
| | $\tau$ (kPa) | $\sigma$ (kPa) | FS | $\tau$ (kPa) | $\sigma$ (kPa) | FS | $\tau$ (kPa) | $\sigma$ (kPa) | FS | $\tau$ (kPa) | $\sigma$ (kPa) | FS | $\tau\tau$ (kPa) | $\sigma$ (kPa) | FS |
| 70 | 62.55 | 58.17 | 1.075 | 61.55 | 55.83 | 1.103 | 59.41 | 53.21 | 1.117 | 58.56 | 51.06 | 1.147 | 57.32 | 48.51 | 1.182 |
| 72 | 57.30 | 54.22 | 1.057 | 63.49 | 58.40 | 1.087 | 63.58 | 57.87 | 1.099 | 64.51 | 57.28 | 1.126 | 64.48 | 55.73 | 1.157 |
| 74 | 65.14 | 62.96 | 1.035 | 66.30 | 62.49 | 1.061 | 63.78 | 59.44 | 1.073 | 62.74 | 56.97 | 1.101 | 61.26 | 54.04 | 1.134 |
| 76 | 64.42 | 63.04 | 1.022 | 66.27 | 63.24 | 1.048 | 66.00 | 62.27 | 1.060 | 64.86 | 59.65 | 1.087 | 64.92 | 58.09 | 1.118 |
| 78 | 67.38 | 67.52 | 0.998 | 66.16 | 64.62 | 1.024 | 65.33 | 63.06 | 1.036 | 64.22 | 60.39 | 1.064 | 62.67 | 57.23 | 1.095 |
| 80 | 62.15 | 63.99 | 0.971 | 61.18 | 61.23 | 0.999 | 59.06 | 58.11 | 1.016 | 58.23 | 55.63 | 1.047 | 66.44 | 61.46 | 1.081 |
| 82 | 65.20 | 68.25 | 0.955 | 64.09 | 65.27 | 0.982 | 61.75 | 61.88 | 0.998 | 60.79 | 59.19 | 1.027 | 59.43 | 56.04 | 1.060 |
| 84 | 71.73 | 76.39 | 0.939 | 70.32 | 73.02 | 0.963 | 67.48 | 69.18 | 0.975 | 66.28 | 66.14 | 1.002 | 64.61 | 62.57 | 1.033 |
| 86 | 70.16 | 75.98 | 0.923 | 68.82 | 72.55 | 0.949 | 72.60 | 75.53 | 0.961 | 74.28 | 78.95 | 0.941 | 69.24 | 68.28 | 1.014 |
| 88 | 64.65 | 71.35 | 0.906 | 63.56 | 68.07 | 0.934 | 69.25 | 72.93 | 0.950 | 73.63 | 75.52 | 0.975 | 71.55 | 71.41 | 1.002 |

In both the saturated and unsaturated conditions of the slope, the variation of the shear stress and shear strength are related. Similarly, with changing slope angle, both these values were also related. All these values underwent SPSS to analyze correlations between all these parameters. It is noted that almost all soil parameters are closely related, and they depend on each other. In future work, other soil parameters could also be considered to identify correlations. Furthermore, it will be an easy way to use these correlations in any design work or analysis. Table 7 presents the correlations developed using SPSS for all the ten types of material used in the case of an unsaturated seismic analysis.

**Table 7.** Correlations in the case of clay—unsaturated seismic case.

| Material Number | Shear Strength (kPa) |
|---|---|
| 1 | $54.625 - 0.714*\beta + 0.911*\sigma$ <br> $R^2 = 99.8\%$ |
| 2 | $52.408 - 0.660*\beta + 0.906*\sigma$ <br> $R^2 = 99.8\%$ |
| 3 | $51.614 - 0.650*\beta + 0.916*\sigma$ <br> $R^2 = 99.8\%$ |
| 4 | $49.580 - 0.606*\beta + 0.913*\sigma$ <br> $R^2 = 99.8\%$ |
| 5 | $49.891 - 0.645*\beta + 0.980*\sigma$ <br> $R^2 = 99.6\%$ |
| 6 | $49.295 - 0.631*\beta + 0.992*\sigma$ <br> $R^2 = 99.7\%$ |
| 7 | $48.733 - 0.598*\beta + 0.988*\sigma$ <br> $R^2 = 99.5\%$ |
| 8 | $47.371 - 0.573*\beta + 0.990*\sigma$ <br> $R^2 = 99.8\%$ |
| 9 | $46.726 - 0.432*\beta + 0.841*\sigma$ <br> $R^2 = 98.6\%$ |
| 10 | $46.128 - 0.550*\beta + 1.036*\sigma$ <br> $R^2 = 99.6\%$ |

From Table 7, the final correlation is calculated by computing the average value, which is:

$$\tau \text{ (kPa)} = 49.637 - 0.606\beta \text{ (deg.)} + 0.947\sigma \text{ (kPa)} \tag{10}$$

the average $R^2$ comes out to be 99.6%.

## 4. Discussions

Slope stability analysis is one of the most challenging fields in geotechnical engineering. Landsliding causes huge losses to human lives as well as the economy of a country. Normally slope stability analysis is performed in conventional ways, but as soil behavior is not homogenous, it is strongly recommended to analyze soil slopes using theoretical as well as practical approaches. Without the collection of samples and testing in a laboratory, it will not give accurate results. This research strongly recommends collecting soil samples from the required site, testing in a laboratory to compute the soil properties and then running the analyses. Furthermore, study the slope in all aspects, such as analyze it in a saturated condition, an unsaturated condition and apply dynamic loads as well. Moreover, other soil types also need to be considered in future work as fissured clays have a different response compared to non-fissured clays [36].

Using Equations (9) and (10), the shear strength for any saturated and unsaturated slopes, respectively, can be computed, provided that the material properties are in the range of Table 1. For example, if the slope angle is 74 degrees and the shear stress is 49 kPa, then the shear strength for this specific case in a saturated condition will be approximately 32.22 kPa, and in an unsaturated condition, it will be 51.20 kPa. The factor of safety can be calculated using Equation (11) as follows:

$$\text{Factor of safety} = \text{Shear strength}/\text{Shear stress} \tag{11}$$

Factor of safety in a saturated condition = 32.22/49 = 0.66,

Factor of safety in an unsaturated condition = 51.20/49 = 1.04,

This equation can be extended to find the correlation of the same parameters while considering the horizontal and vertical seismic coefficients together. Although the vertical seismic effect in most of the cases is negligible, it affects the slope in some cases. Normally shear strength is computed using Coulomb's equation:

$$\tau = c + \sigma \tan \Phi \tag{12}$$

Combining Equation (12) with 9 and 10, a new correlation is developed as follows.

$$c + \sigma\tan\Phi = 30.482 - 0.385\beta + 0.617\sigma = 49.637 - 0.606\beta + 0.947\sigma \tag{13}$$

All these developed correlations are not generalized equations that can be applied in all scenarios. These correlations can be applied only in specific conditions; the material properties must be in the range of Table 1.

All other discussions regarding this research are mentioned in the result section, with each step of the study.

## 5. Conclusions

Soil slopes in two different conditions such as saturated and unsaturated conditions, are analyzed considering the horizontal seismic coefficients. Two hundred types of material properties were considered, and finally, correlations between shear strength, slope angle and shear stress are developed as Equations (9) and (10).

- From Equations (9) and (10), it is clear that the higher the value of $\beta$ and $\sigma$, the value of $\tau$ will be lower. The relationship could be checked for any slope stability case. For any other slope stability project, the shear strength can be obtained by simply applying Equations (9) and (10). The factor of safety can also be obtained by dividing shear

strength by shear stress. These equations can be used to compute the shear strength of any soil slope in the given material properties range.

- The applicability of Equations (9) and (10) is above 99 percent.
- The same equations can be extended to other material types with the same procedure of analysis. If a material's properties exist between the specified values, then the shear strength can be calculated with interpolation. The same work can be extended to analyze the seismic condition by also considering the vertical seismic coefficient to get a clearer idea of the shear strength variation in complex conditions.
- In future work, the correlation of shear strength, factor of safety and slope angle with the variation of the width of an embankment will be studied.

**Author Contributions:** Conceptualization, investigation, methodology, writing original draft by M.I.K. and resources, supervision, funding by S.W. All authors have read and agreed to the published version of the manuscript.

**Funding:** This work is conducted with support from the National Natural Science Foundation of China (Grant Nos. U1602232 and 51474050), Doctoral Scientific Research Foundation of Liaoning Province (Grant No. 20170540304 and 20170520341), China Scholarship Council (Grant No. 201806080103), Key Research and Development Program of Science and Technology in Liaoning Province, China (Grant No. 2019JH2/10100035) and the Fundamental Research Funds for the Central Universities (Grant No. N170108029).

**Institutional Review Board Statement:** Not applicable.

**Informed Consent Statement:** Not applicable.

**Data Availability Statement:** Not applicable.

**Acknowledgments:** The authors would like to thank the Applied Sciences (MDPI) journal for considering this paper for publication. The authors are also grateful to Northeastern University China for supporting this work.

**Conflicts of Interest:** The authors declare no conflict of interest.

## Nomenclature

| | |
|---|---|
| FS | Factor of safety |
| $\tau$ | Shear strength of soil |
| $\beta$ | Slope angle |
| $\sigma$ | Shear stress of soil |
| c | Cohesion of soil |
| $\Phi$ | Soil friction angle |
| SPSS | Statistical package for social sciences (Software) |
| SI | System international |

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
