# Peer review of "Slope Stability Analysis to Correlate Shear Strength with Slope Angle and Shear Stress by Considering Saturated and Unsaturated Seismic Conditions"

_applsci, doi:10.3390/app11104568_

Round 1
Reviewer 1 Report
The article contains an interesting analysis of slope stability under various load conditions. Below are some comments and suggestions.
- Line 22, the beginning of the sentence should be corrected.Greek characters should appear later in the sentence, not at the beginning.
- Figure 1, write down what stresses are represented by the two black arrows. Moreover, due to the fact that it is a model, the side of the applied load should be additionally presented.
- In the Chapter 2.1, describe what equipment was used to take the samples (drill diameter, drive).
- Lines 115-116, sentence “… .Such as triaxial test, moisture content test, direct shear test etc….” In this sentence, the wording etc should be avoided, it is incorrect. Describe exactly what these tests were. Moreover, it is necessary to describe the load velocity in the triaxial test, describe this apparatus and write what formulas were used to determine the shear strength, cohesion and friction angle.
- Table 1, add units for the friction angle.
- Figure 6, write what HSC stands for.Also, the sign of the angle and the arrow are incorrect.Use the markings in accordance with the technical drawing.
- Figure 7, in the rectangle no 1, write what these parameters are; avoid wording etc. Rectangle no 3 should be described as "... Find seisimc shear strength and shear stress for each material type .." which means, by what methods, badan seismic shear strength and shear stress are found. Is it about dynamic shear strength and stress?
- Line 138 "… .Similarly The…", please correct the sentence.
- Line 150 “… .Table 1 shows the material properties. b range is kept Seventy (70) to Eighty .. ”please correct your mind.
- Line 151, "Sixty (60) to Ninty", is not capitalized.
- Line 155 ”…. methods, Ordinary… ”, please correct the sentence.
- Regarding the Statistical Package for the Social Sciences (SPSS) software, what statistical analyzes have been performed (correlation, variance, regression… ..) ?.
- Line 156, should be written in a few sentences, what characterizes the Fellenius analysis method. 14. Line 158, figure 5, should have a different numbering, rather 8. Moreover, this drawing should be labeled 8a and 8b. For this drawing, markings for all values ​​(b, kw, w, EN, TN, S, sl, F, L, P, pl… ..) should be written. Formula no. 4, the parenthesis is missing. In addition, you should write what is the difference between arms: xi, Ri, fi and ei.
- In the Chapter 3.1 it is necessary to describe in detail how the horizontal seismic was modeled. First of all, write what the value 0.3 means in relation to the horizontal seismic coefficient. In Figure 6, it should be indicated in which layers the shock source is, at what distance and what is the model's damping. In addition, the drawing should have a different numbering.
- Line 190, add the unit to the value 16 and 0.16; furthermore correct the sentence "... Varies ..". 17. In the table 2, please write down what df means;F;Sig;B, t. Moreover, please write in two or three sentences why the ANOVA analysis of variance statistical method was chosen (how many factors acting simultaneously were taken into account).
- The designation of the β angle in Figure 7 should include the known angle. Renumber the figure. 19. Tables 3-4; 6-7, add units to the designation of individual stresses…. degrees and write the designation to M.
- Figure 8, write the marking to M.
- Formula no 8 and 9, add a unit.
- Figure 10, the color scale for the safety factor does not match the colors in some figures, for example FS = 0.866 or FS = 0.825. Adjust the scale.
- Line 279, correct KPa entry.
- Lines 283-285 add commas at the end of the sentence.
- Line 291, write what new correlation can be developed.
- In the suggestion, refer to the seismic events analyzed in the example.
Author Response
I am very thankful to reviewer 1, especially for this statement of reviewer:
“The article contains an interesting analysis of slope stability under various load conditions.”
Below are comments and suggestions of reviewer and all these comments and suggestions are answered accordingly.
- Line 22, the beginning of the sentence should be corrected. Greek characters should appear later in the sentence, not at the beginning.
Answer: Corrected. Check line 22 in revised version.
- Figure 1, write down what stresses are represented by the two black arrows. Moreover, due to the fact that it is a model, the side of the applied load should be additionally presented.
Answer: Title specified to the arrows and side force. Check figure 1 on page 3 in revised version.
- In the Chapter 2.1, describe what equipment was used to take the samples (drill diameter, drive).
Answer: Soil samples are collected by using drill machine having 6 inches diameter.
(Mentioned in section 2.1). Check line 111 and 112 in revised version.
- Lines 115-116, sentence “Such as triaxial test, moisture content test, direct shear test etc….” In this sentence, the wording etc. should be avoided, it is incorrect. Describe exactly what these tests were. Moreover, it is necessary to describe the load velocity in the triaxial test, describe this apparatus and write what formulas were used to determine the shear strength, cohesion and friction angle.
Answer: Word “etc.” is removed. Mohr-Coulomb envelope criteria is used to compute the cohesion and friction of soil. The general form of Mohr-Coulomb criteria is shown as equation 1.
t = c + stanf
Figure 6 presents the Mohr-Coulomb envelope. Check 125 to 130 in revised version.
- Table 1, add units for the friction angle.
Answer: Friction angle unit is degrees which is mentioned now. Check table 1 on page 6 in revised version.
- Figure 6, write what HSC stands for. Also, the sign of the angle and the arrow are incorrect. Use the markings in accordance with the technical drawing.
Answer: HSC stands for Horizontal Seismic Coefficient and the arrow is removed. Figure 6 becomes figure 7 now as one extra figure is added. Check figure 7 on page 6 in revised version.
- Figure 7, in the rectangle no 1, write what these parameters are; avoid wording etc. Rectangle no 3 should be described as "Find seisimc shear strength and shear stress for each material type" which means, by what methods, badan seismic shear strength and shear stress are found. Is it about dynamic shear strength and stress?
Answer: Word “etc.” is removed. Yes, it is dynamic shear strength and shear stress. Word “seismic” is replaced by dynamic. And the slope factor of safety values are computed by using limit equilibrium method i.e. Fellenius method. Check figure 8 on page 7 in revised version.
- Line 138 "Similarly The…” please correct the sentence.
Answer: Corrected. Check line 147 in revised version.
- Line 150 “Table 1 shows the material properties. b range is kept Seventy (70) to Eighty .. ”please correct your mind.
Answer: Corrected.
- Line 151, "Sixty (60) to Ninty", is not capitalized.
Answer: Corrected. Check line 159 in revised version.
- Line 155 ”…. methods, Ordinary… ”, please correct the sentence.
Answer: Corrected and word ordinary is removed. Check line 165.
- Regarding the Statistical Package for the Social Sciences (SPSS) software, what statistical analyzes have been performed (correlation, variance, regression… ..) ?.
Answer: Linear regression analysis is performed using SPSS linear regression analysis software.
- Line 156, should be written in a few sentences, what characterizes the Fellenius analysis method.
Answer: Fellenius analysis method is considered as it gives smaller value for slope factor of safety. The limit equilibrium software namely Slide offers all analysis methods such as Bishop simplified, Corps of Engineers # 1, Corps of Engineers # 2, GLE / Morgenstern-Price, Janbu simplified, Janbu corrected, Lowe-Karafiath, Ordinary/Fellenius and Spencer method. Out of all these methods, it is observed that Fellenius method gives lower slope factor of safety. And considering lower value for slope factor of safety while designing a slope provides more factor of safety. Therefore Fellenius method is preferred in this analysis. Check line 165 – 173 in revised version.
- Line 158, figure 5, should have a different numbering, rather 8. Moreover, this drawing should be labeled 8a and 8b. For this drawing, markings for all values ​​(b, kw, w, EN, TN, S, sl, F, L, P, pl… ..) should be written. Formula no. 4, the parenthesis is missing. In addition, you should write what is the difference between arms: xi, Ri, fi and ei.
Answer: The figure is divided into two parts, such as 9a and 9b as suggested by reviewer. All the parameters defined in figure 9b as suggested. Check on page 8 in revised version.
- In the Chapter 3.1 it is necessary to describe in detail how the horizontal seismic was modeled. First of all, write what the value 0.3 means in relation to the horizontal seismic coefficient. In Figure 6, it should be indicated in which layers the shock source is, at what distance and what the model’s damping is. In addition, the drawing should have a different numbering.
Answer: The horizontal seismic load was applied using seismic loading option in Slide software. The Seismic Coefficients are dimensionless coefficients that represent the maximum earthquake acceleration as a fraction of the acceleration due to gravity. Typical values are in the range of 0.1 to 0.3. In this paper, maximum value for horizontal seismic coefficient is considered as 0.3. Check line 204-209 in revised version.
The shock source details can be judged from shear strength graphs provided as figure 12a and 12b. Factor of safety value is taken at a distance of maximum shear strength shown in figure 12a and 12b. 13% damping difference is observed between seismic and nonseismic analysis. Figure 6 is corrected as figure 10 now. Check figure 10 on page 9 in revised version.
- Line 190, add the unit to the value 16 and 0.16; furthermore correct the sentence "... Varies ..".
Answer: Unit is added that is “m”. The model coordinates are in meters. Check line 212 where it is written “The coordinates (m)….”
In the fifth coordinate, varies with slope angle means that this coordinate value changes with the variation of slope angle from 70 to 88 degrees. Word Varies is written as varies now. V corrected. Check line 212.
- In the table 2, please write down what df means; F; Sig; B, t. Moreover, please write in two or three sentences why the ANOVA analysis of variance statistical method was chosen (how many factors acting simultaneously were taken into account).
Answer: Table 2 is removed as suggested by reviewer 2. It was just to show the linear regression summary. The only important part of this table was the coefficients part which is taken already used in the correlations.
- The designation of the β angle in Figure 7 should include the known angle. Renumber the figure.
Answer: The b angle mentioned is in degrees. FS and b both are mentioned at every slope analysis. The figure number is also corrected as figure 11. Check figure 11 on page 10.
- Tables 3-4; 6-7, add units to the designation of individual stresses…. degrees and write the designation to M.
Answer: Units designated to all parameters. Check table 3 and 4 on page 11 and table 6 and 7 on page 14.
- Figure 8, write the marking to M.
Answer: Figure 8 is now corrected as figure 12a and 12b. M is designated and corrected. Check figure 12a and 12b on page 11 and 12 respectively. Also check line 238 to 241 in revised version.
- Formula no 8 and 9, add a unit.
Answer: Units added. And formulas 8 and 9 numbers become 9 and 10 in the revised version. Check line 253 and 287.
- Figure 10, the color scale for the safety factor does not match the colors in some figures, for example FS = 0.866 or FS = 0.825. Adjust the scale.
Answer: The figure is replaced and corrected. Figure 10 becomes figure 11 in the revised version. Check figure 11 on page 10.
- Line 279, correct KPa entry.
Answer: Corrected. Check line 303.
- Lines 283-285 add commas at the end of the sentence.
Answer: Commas added. Check line 307-309.
- Line 291, write what new correlation can be developed.
Answer: c + stanf = 30.482 - 0.385b + 0.617s = 49.637 - 0.606b + 0.947s
This equation is developed and mentioned in paper now. Check line 316.
- In the suggestion, refer to the seismic events analyzed in the example.
Answer: Right.
Once again, I pay thanks to reviewer 1 for your nice comments and suggestions.
Muhammad Israr Khan
Corresponding Author

Reviewer 2 Report
All comments are reported to the attached file.

Author Response
I am very thankful to reviewer 2, for all the nice comments and suggestions.
Below are comments and suggestions of reviewer and all these comments and suggestions are answered accordingly.
- Lines 31-38: this part of the introduction is correct, but many of the reported sentences were already investigated in previously published papers. Therefore, several references need to be added here. Some suggested references are reported at the end of this report.
Answer: The suggested four references are really useful and matching to this paper. They are cited in the introduction of this paper. Check line 34-36 in revised version.
- Line 63: a scientific paper should be written entirely in third person. At this line, the Authors use the word ‘we’, that should be avoided. This comment is valid for the whole manuscript, that the Authors are encouraged to carefully check.
Answer: Word “we” is removed and it was only at one place. Check line 64 and 65 in revised version.
- It seems that Figs. 1 and 2 are not from the authors. If this is the case, the source should be cited.
Answer: Figure 1 belongs to author and figure 2 reference is mentioned as Ref. 23. Check line 94 in revised version.
- Line 102: change ‘has’ in ‘have’
Answer: Changed. Check line 104.
- Quality of Fig. 3 should be increased. Furthermore, this figure should be integrated with the indication of the location of the site in a wider area.
Answer: Quality is improved now and figure 3 is replaced with a broader view as suggested. Check figure 3 on page 4.
- Be careful, there are two different fig. 5 and two different fig. 6. Please check the figures’ numbers.
Answer: All figure numbers are revised and corrected. Total number of figures are 14. Figure 9 and 12 are divided into two figures, such as 9a, 9b and 12a, 12b respectively.
- Figs 4 and 5 (line 118) are not strictly required. Soil samples and triaxial tests are worldwide well known, and there is no reason to show these pictures in an original paper.
Answer: There are many virgins of almost all the laboratory equipments. These figures are added to specify it for better understanding. Especially for new comers who are not well familiar with these equipments.
- Line 113: the Authors write that 20 samples were collected and subjected to laboratory tests. These results are reported in table 1, which however contains only 10 results. Why there is this difference?
Answer: It is corrected as 10 number of boreholes. Check line 117 in revised version.
- Table 1 (and elsewhere in the manuscript): more attention has to be paid to the units: the cohesion is measured in kPa (lowercase k, capitol P, lowercase a; i.e. kPa, not KPa). The same for the unit weight (kN/m3 rather than KN/m3)
Answer: Correction made and units are written as suggested. Units corrected as kPa and kN/m3 throughout the paper.
- Lines 126-130: does the slope shown in Fig. 6 (line 132) refer to the site shown in Fig. 3? In the current version of the manuscript this is not clearly stated and could generate confusion.
Answer: Yes, the slope drawn refers to figure 3. It is mentioned in line 135 that:
“According to the actual site condition, a 2D slope was modelled in Slide (2D limit equilibrium software) for further analysis.”
- Line 154: the unit of the friction angle has to be included.
Answer: Unit included now. Check line 163 and 164 in revised version.
- One of the main criticism of the paper is that the method of Fellenius is employed. It is well known that this method underestimates the factor of safety. This underestimation is stronger when the depth of the slip surface increases and when the slip surface is circular. On the contrary, it provides better results when the slip surface is a plane. Furthermore, the inaccuracy of this method increases for high values of the pore pressure. Finally, the equilibrium of the horizontal forces is neglected in this method. This aspect makes problematic the employment of this method when horizontal forces are also considered, such as the seismic one. It is clear that for the application presented in this study the method of Fellenius is not the most suitable one. In particular, this Reviewer believes that the current version of the manuscript is not persuading in the current form for researchers that are into the field of geotechnical engineering. In order to confer a better impression, a more suitable method should be employed to perform this application. Obviously, the method of Bishop would be much better, but analogously to that of Fellenius, horizontal forces are ignored. Since the seismic actions are also accounted for, this Reviewer would suggest to employ a rigorous method, such as Morgenstern and Price or Spencer.
Answer: Fellenius analysis method is considered as it gives smaller value for slope factor of safety. The limit equilibrium software namely Slide offers all analysis methods such as Bishop simplified, Corps of Engineers # 1, Corps of Engineers # 2, GLE / Morgenstern-Price, Janbu simplified, Janbu corrected, Lowe-Karafiath, Ordinary/Fellenius and Spencer method. Out of all these methods, it is observed that Fellenius method gives lower slope factor of safety. And considering lower value for slope factor of safety while designing a slope provides more factor of safety. Therefore Fellenius method is preferred in this analysis. It is same like if a structure can support 10 ton of load but we consider that it support 9 ton or 8 ton. It is just to be on safe side. Same criteria is followed while choosing Fellenius method. Although the software gives results for all other methods which are suggested by reviewer. Details added in line 165-173 in revised version.
- Fig 5 (line 158) is awful. This figure should be drawn in a more professional way, and the usage of a professional drawing tool, such as for example Photoshop, would be required to make figures that will be published on an international journal. This comment should be extended also to all the other figures.
Answer: The figure is replaced and quality improved now. Check figure 9a and 9b on page 8 in revised version. All other figures are also improved.
- Line 170 (Eq. 4): there is a bracket that is open but not closed.
Answer: Corrected. Bracket closed now. Check line 188. Equation 4 becomes equation 5 in the revised version.
- Lines 159-177: the method of Fellenius is well known among geotechnical engineers since several decades ago. Therefore, this Reviewer believes that there is no reason to demonstrate how the factor of safety is calculated. The only indication of Eq. 7 would be enough.
Answer: Yes the point of reviewer is 100% valid but this method is explained a little bit in few lines for beginners or student level people. Because readers of the paper may include low, medium and high level of conceptual people.
- Line 204: the sense of table 2 is not clear. Please, clarify better what this table means or delete it.
Answer: Table 2 is removed as suggested by reviewer. It was just to show the linear regression summary. The only important part of this table was the coefficients part which is taken already in the correlations.
- In tables 3 and 4, the values of the factor of safety should be included as well (the same comment should be followed also for the unsaturated soils).
Answer: Factor of safety values are added to the tables as suggested. Check table 3 and 4 on page 11 and table 6 and 7 on page 14 respectively in the revised version.
- Section 3.2: this section (contrarily to the previous one) requires an explanation about how the factor of safety is evaluated. Specifically, what method did the Author use to calculate FS? What criterion did they use to model the shear strength of the unsaturated soil on the slip surface? How did they account for the soil suction? And what value of suction did they use?
Answer: In this section, same method (Fellenius) is followed for slope stability analysis, computing shear strength, shear strength and factor of safety. Measurement of soil suction is done through Slide software. It is measured in terms of the height of the water column (h) suspended in the soil.
- Line 279: please write ‘kPa’
Answer: Corrected as “kPa”. Check line 303.
Once again, I pay thanks to reviewer 2 for your nice comments and suggestions.
Muhammad Israr Khan
Corresponding Author

Round 2
Reviewer 2 Report
The Authors have properly addressed the issues raised by this Reviewer.
Author Response
Response to Reviewer 2
The manuscript is revised and checked through an English expert. All grammatical mistakes are corrected and now the revised version is free from any grammar mistake.
Muhammad Israr Khan
Corresponding Author
